# Acid hydrolysis of saponins extracted in tincture

**Jamie Love** [1]*, **Casey R. Simons** [2]

**1** Synapses, Millbrae Cottage, Mill of Fyall, Alyth, Scotland, United Kingdom, **2** Department of Chemistry and Biochemistry, Utah State University, Logan, Utah, United States of America

* drjamielove@synapses.co.uk

## Abstract

### Background

Saponins are secondary metabolites from plants added to shampoos and beverages to make them foam, and the sapogenins released from them upon acid hydrolysis are commonly used as starting materials for steroidal drugs. However, current methods embed the saponin in a thick "gum" material consisting of multiple impurities. This gum limits access to the saponin, reducing the efficiency of hydrolysis and requiring large amounts of heat, organic solvents and effort to recover the sapogenin. For centuries, herbalists have been making tinctures by soaking plant materials at room temperature, in mixtures of alcohol and water. Many herbal tinctures contain saponins floating freely in solution, gum free. The saponin from sarsaparilla (*Smilax* spp), sarsasaponin, yields the sapogenin, sarsasapogenin, upon acid hydrolysis. The retail price of sarsasapogenin is very high but would be lower if the "gum problem" could be avoided.

### Materials and methods

We incubated sarsaparilla tincture under different conditions of temperature, acidity and duration then used quantitative nuclear magnetic resonance (qNMR) to measure the amount of sarsasapogenin produced by hydrolysis as well as the amount of its epimer, smilagenin.

### Results and discussion

Most, if not all the sarsasaponin in sarsaparilla root powder is extracted into a solution of 45% ethanol (55% water) at room temperature and stays suspended without formation of any particles (gum). Acid hydrolysis of the saponin in this solution is very efficient, approaching 100%. The sarsasapogenin released by hydrolysis and the smilagenin produced by its epimerisation, migrate into the chloroform phase.

### Conclusion

Sarsaparilla saponin diffuses into and disperses in a solution of alcohol:water (45:55) at room temperature. Hydrolysis of saponins in tincture provides a simple, inexpensive and environmentally friendly alternative.

**Data Availability Statement:** All relevant data are within the manuscript.

**Funding:** The author(s) received no specific funding for this work.

**Competing interests:** The authors have read the journal's policy and declare the following competing interests: Synapses is a self-funded personal laboratory created and owned by the corresponding author. There are no patents, products in development or marketed products associated with this research to declare. This does not alter our adherence to PLOS ONE policies on sharing data and materials.

# Background

## Saponins and sapogenins

Saponins are amphipathic molecules found in a variety of plants that, when shaken in aqueous solutions, produce a persistent soap-like foam (from which their name is derived. Latin "sapo"). Saponins are amphipathic because they are composed of a hydrophilic sugar moiety, called a glycoside or glycone, and a lipophilic steroidal moiety, called a sapogenin or aglycone, joined through an O-linked glycosidic bond. This bond can undergo hydrolysis by acids or by enzymatic/microbial activity. When the bond is broken, the foam disappears, and the glycoside disperses in an aqueous solution. The sapogenin, precipitates from an aqueous solution or migrates to, and disperses throughout, an organic solvent.

Sarsasaponin (Chemical Abstracts Service Number, CAS# 19057-61-5) is a saponin from sarsaparilla (*Smilax spp*.) with formula $C_{51}H_{84}O_{22}$ (1,048 g/mol).

Sarsasapogenin (CAS# 126-19-2) is the lipophilic moiety released by the hydrolysis of sarsasaponin's glycosidic bond. It is a steroid (β-hydroxy-5β-spirostane) with formula $C_{27}H_{44}O_3$ (416.6 g/mol). See Fig 1.

Smilagenin (CAS# 126-18-1) is an isomer of sarsasapogenin, distinguished by a stereocenter at C25 as shown in Fig 2.

Both molecules have additional stereocenters, at C-20 and C-22, so smilagenin and sarsasapogenin are just two of the eight possible isomers in this stereoisomer family. The configurations for smilagenin and sarsasapogenin are identical at both of these additional stereocenters (described as 20S and 22R), therefore, smilagenin and sarsasapogenin are epimers, differing at a stereocenter, but they are not enantiomers because they possess additional stereocenters. Smilagenin and sarsasapogenin are diastereomers. Unlike enantiomers, diastereomers can, and often do, have different physical properties due to differences in their spatial arrangements. Table 1 displays some examples of these properties for smilagenin and sarsasapogenin.

Reactions catalyzed by strong acids lead to the formation of the most stable product among the eight possible stereoisomers. The driving force pushing the reaction to become smilagenin is the steric strain release accompanying the isomerization of 25S to 25R. The reverse transformation is possible but the content of the 25S isomer, sarsasapogenin, in the equilibrium mixture is expected to be low [5].

To synthesise these compounds, one must consider how each step affects the configuration at all three stereocenters. Such stereospecific synthesis is difficult to control, so a heterogenous product usually results, requiring purification by a series of crystallisations, chromatography, distillations, etc. Therefore hemisynthesis (the synthesis of a new compound derived from an existing natural product) from available stock materials, commonly diosgenone, is expensive.

## Saponins and sapogenins from plants

In 1934 Jacobs and Simpson described the isolation of sarsasapogenin from the roots of sarsaparilla *Smilax ornata* Hook (Smilacaceae) 1889 (not Lem. 1865) [6]. Two years later, Askew, Farmer and Kon isolated smilagenin from the same source, preparing the saponin and hydrolysing it as described by Jacobs and Simpson, but using different means to purify it [7]. Their protocol can be summarised in these three steps.

**Prepare the saponin for hydrolysis.**   *Smilax ornata* root powder was percolated with cold 95% denatured alcohol, and the alcoholic extract was concentrated, under reduced pressure, to a "brown, viscous gum". To "defat" this material, it was extracted three times using "lignon 80-90°" resulting in a "fat-free gum", which Jacobs and Simpson describe as a "brown, indistinctly crystalline solid".

## Sarsasaponin (a saponin)

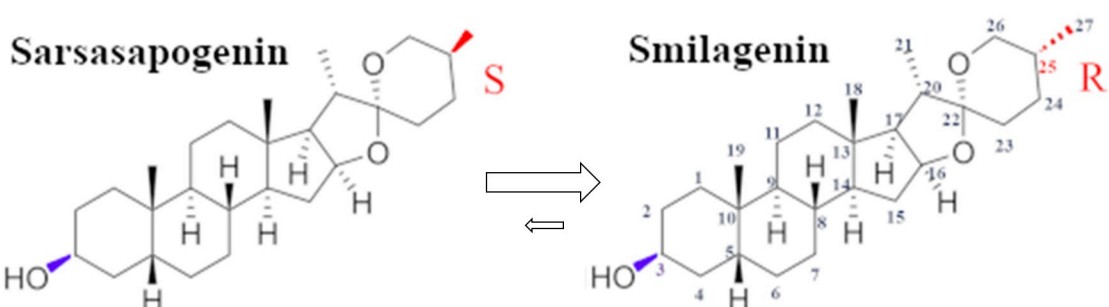

**Fig 1. Sarsasaponin.** Sarsasaponin is composed of a hydrophilic sugar moiety (a glycoside, coloured blue) and a lipophilic steroidal moiety (a sapogenin, coloured black) joined through a glycosidic bond (purple) to make an amphipathic molecule that produces a persistent foam when shaken in water.

**Fig 2. Fisher projections of smilagenin and sarsasapogenin.** The smilagenin molecule, on the left, displays the numbering system of each carbon atom. The glycoside was attached at the C3 position but after hydrolysis is occupied by a hydroxyl group (here, attached by a purple bond). At the other end of the rings is C25, a chiral atom bound to a methyl group (C27) in the R-configuration for smilagenin or the S-configuration for sarsasapogenin (in red).

**Table 1. Some physical properties of smilagenin and sarsasapogenin.**

| Diastereomers | Melting Point (°C) [1] | crystals in acetone [2] | Solubility [3] | Solubility in Ethanol [4] |
|---|---|---|---|---|
| Sarsasapogenin 20S 22R 25S | 200.5 | long, prisms, needles | soluble in ethanol, acetone, benzene & chloroform | 1 mg/ml (2.40 mM) |
| Smilagenin 20S 22R 25R | 185 | Needles | very soluble in acetone, benzene, & ethanol | ≥ 10 mg/ml (24.00 mM) |

**Hydrolyse the saponin.** They hydrolysed their crude saponin preparations of "fat-free gum" by incubating one-kilogram portions (bigger portions were too difficult to manipulate), in 1.2N HCl at 80 °C for 45 minutes. After collecting the precipitant by centrifugation, it was refluxed (78 °C) for 1½ hours in 1.2N HCl and 95% ethanol "to complete the hydrolysis".

**Collect the sapogenin.** At this point, the teams chose different ways to clean up their preparation, ending up with different diastereomers.

Jacobs and Simpson conducted a series of crystallisations in ethanol with ammonia hydroxide, followed by extractions with acetone containing "black-bone" (a porous, black, granular material produced from charred animal bones), then extracted with ether in a Soxhlet apparatus, before the final crystallization in absolute acetone—producing sarsasapogenin crystals that melted at 199–199.5 °C with a yield of 0.2% (450 g of sarsasapogenin from 225 kg of plant).

Askew, Farmer and Kon repeatedly extracted with benzene and water. The benzene was passed through a long column of active alumina, then fractionally crystallised in absolute acetone—producing smilagenin crystals that melted at 183–184 °C with a yield of 0.02%.

By the middle of the twentieth century saponin and sapogenin chemistries were well established but isolation and hydrolysis of saponin was still stuck with the "gum problem". In 1952 Monroe E Wall (*et al.*) published a paper detailing a protocol for the isolation of crude sapogenins that they had developed from a study of more than a thousand sapogenaceous plant samples [8]. Their method starts with extraction of saponins in hot 85–95% ethanol or isopropanol to produce a gum. The fat-soluble material was removed by extraction in benzene, followed by extraction in butanol, which leaves the saponins embedded in gum. To hydrolyse their gum, hydrochloric acid is added to the aqueous saponin solution to make it 4 N and this acidic mixture is refluxed for 3 to 4 hours.

By the twenty-first century, it was acknowledged that considerable amounts of solvents, heat and effort are needed to prepare crude saponin gum, so greener technologies are now being pursued such as ultrasound-assisted, microwave-assisted, and accelerated solvents. However, "researchers are more inclined to selection of the conventional extraction techniques (70%)" [9].

## Herbology

Herbalists attribute the medicinal properties of many of their tinctures to the foam they produce [10] and over the centuries developed methods to make extracts from plant materials that produce large amounts of foam, that we now understand is caused by saponin. The plant materials, called "marc", are soaked at room temperature in a solution, called "menstruum", which is usually diluted ethanol with a ratio of water to ethanol that is specific to the "medicament". The liquid portion is collected, including the liquid squeezed from the wet marc, and filtered to make "tincture". Saponins in tincture float freely in solution, not encased in gum, so are exposed to aqueous protons, which are the limiting reagent in acid hydrolysis of saponins. Apparently, this has gone unnoticed by organic chemists.

Sarsaparilla tincture is made from the roots of plants in the genus *Smilax*, which comprises about 300 species, worldwide, twenty of which have been reported to have a total of 104 steroidal saponins [11] (however, many of these share the same sapogenin and differ only in the glycoside). A search of Dr Duke's Phytochemical and Ethnobotanical Databases [12] found only limited information about the species, *Smilax ornata*, but a broader search across the genus *Smilax* found the data summarised in Fig 3.

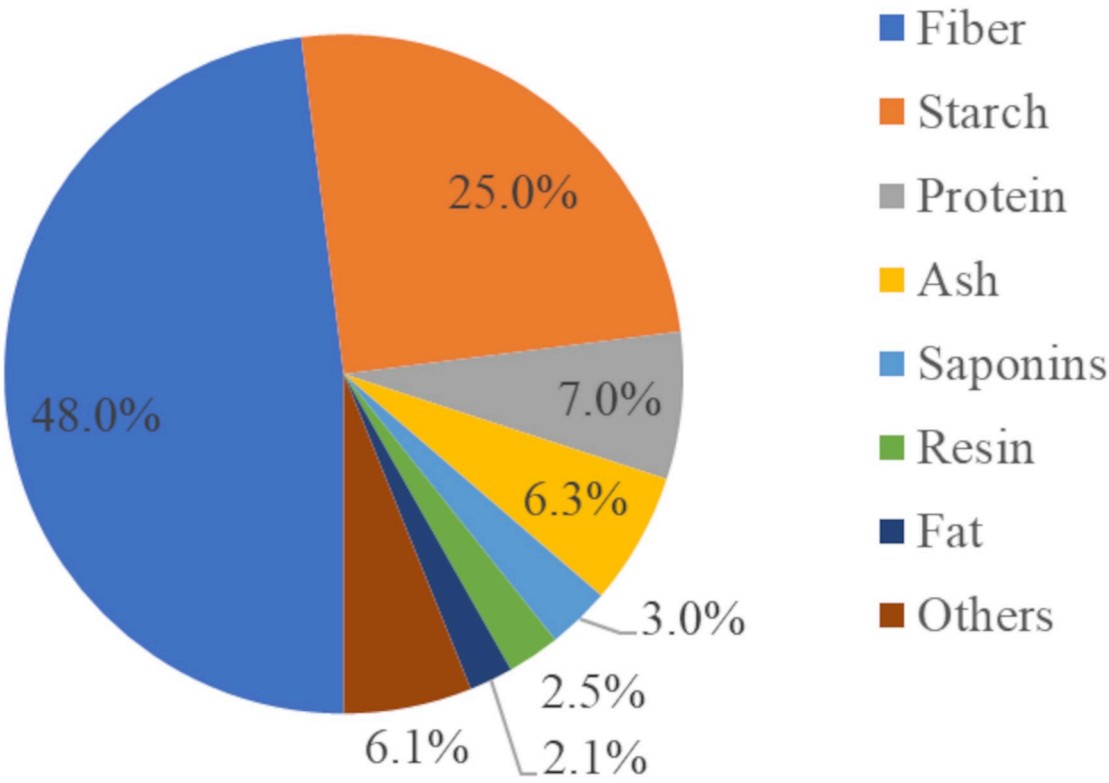

**Fig 3. The composition of *Smilax* spp, root [13].**

As many as six species of sarsaparilla are reportedly used to make tincture but only three are commonly used [14]. *Smilax ornata* has been discussed. *Smilax officinalis* contains saponins that upon hydrolysis yields sarsasapogenin but also neotigogenin and 25S-spirostan-6 beta-ol [15]. These sapogenins do not interest us and would only complicate the extraction. *Smilax aristolochiifolia* contains a saponin that upon hydrolysis yields sarsasapogenin [16], so might offer an alternative source, but we chose to study *S. ornata* because it is better characterised.

## Objective

Our objective was to evaluate sarsaparilla tincture as feedstock for production of sarsasapogenin and smilagenin.

## Materials and methods

### Persistent foam assays

Persistent foam assays, illustrated in Fig 4, were used to monitor the decrease in the saponin.

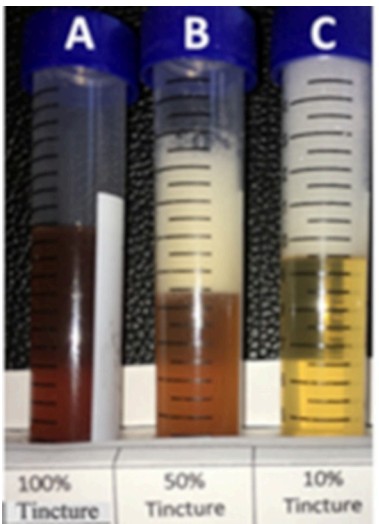
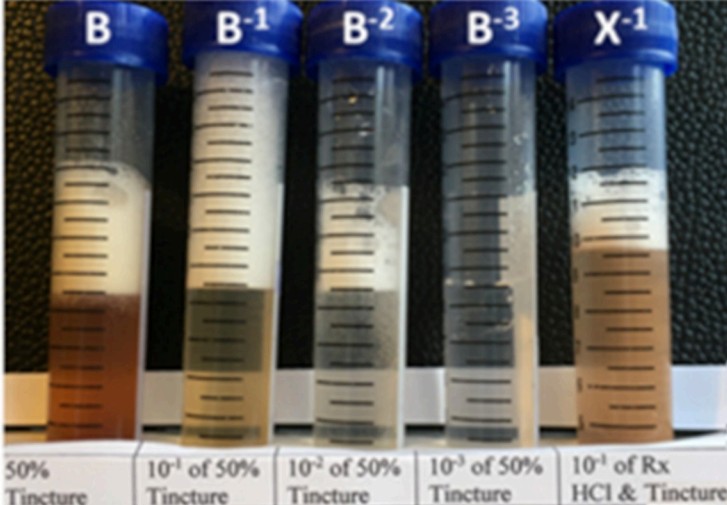

**Fig 4. Photographs of foam assays.** The photograph on the left shows dilutions of tincture after they were shaken vigorously for 15 seconds. The photograph on the right was take 10 minutes after the tubes were shaken. Tube B is the same tube in both photographs, but it has lost about half a millilitre of foam during the 10-minute interval, leaving 3.5mls of persistent foam.

Alcohol interferes with the formation of foam. Pure tincture (100%, A in Fig 5) contains 45% ethanol and cannot make foam. Tincture diluted by half with water (B in Fig 5) is 22.5% ethanol and able to produce about 4 ml of foam. The largest amount of foam is produced when 1ml of tincture is diluted in 9 ml of water (C in Fig 5) which contains only 4.5% ethanol.

To assay for persistent foam, 1ml of the solution (usually Reaction Mix) was added to 9 ml of water in a 15 ml graduated tube. The lid was screwed on tightly and the tube was shaken

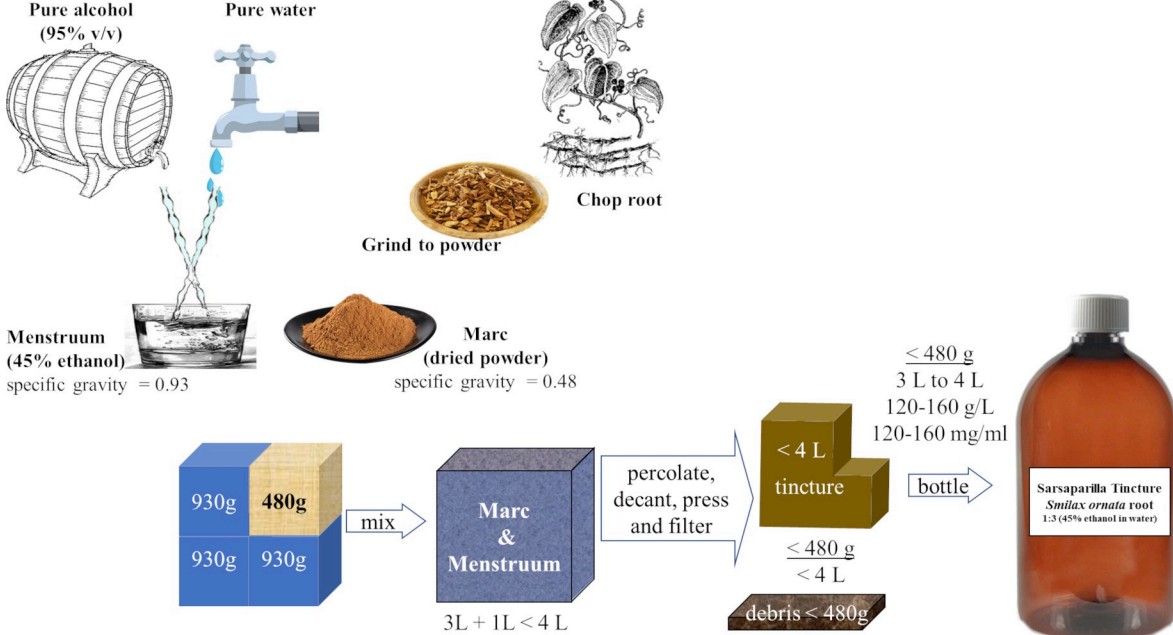

**Fig 5. How our tincture was made.** A volume of marc (plant material) was mixed in 3 volumes of menstruum. In metric units that is 480 grams of powder (specific gravity of 0.48) in 3 litres of 45% ethanol (specific gravity of 0.93). The concentration of marc retained in solution must be determined by measuring the dry weight of the tincture.

vigorously for 15 seconds then set stationary. After 10 minutes a photograph was taken of the foam columns and volumes recorded in millilitres (ml) using the gradations on the side of the tube. If there was plenty of saponin, the airspace was full of persistent foam ($B^{-1}$ in Fig 5). Solution $B^{-2}$ was a $1/10^{th}$ dilution of $B^{-1}$, so represents a hundred-fold dilution of B and produced a measurable amount of persistent ten-fold dilution of (~2.5 ml). Another 10-fold dilution produced $B^{-3}$ which had no persistent foam. Clearly there is a direct relationship between the amount of saponin and the amount of persistent foam, but we failed to find a consistent way to express the relationship quantitatively.

Foam assays are a quick and easy way to screen for the loss of the reagent (saponin), but do not measure the production of product (sapogenin). An overly aggressive reaction may do more than cleave the glycosidic bond, producing unexpected and unwanted products. Therefore, products that retained their persistent foam were scored as "failed hydrolysis" and discarded, while products that lost their ability to produce a persistent foam were prepared for analysis to determine if the missing saponin had been hydrolysed, degraded or both.

### Preparation of saponin for hydrolysis

Sarsaparilla tincture was purchased from "Napiers the Herbalists" of Glasgow [17] who explained that one litre of powdered *Smilax ornata* root (marc), with a density of ~0.48 g/cm$^3$, was immersed in three litres of 45% (v/v) ethanol (menstruum), as illustrated in Fig 5.

This thick suspension was sealed inside a vessel and the menstruum was repeatedly percolated through the marc under pressure. After sufficient percolation (overnight), the liquid was collected, including additional liquid squeezed from the wet marc, and filtered through wine cloths. This liquid, now referred to as "tincture", was aliquoted into brown plastic bottles and hermetically sealed [18].

The *Smilax ornata* root powder used to make the batches of tincture that we used in our experiments, was grown in Mexico and obtained from a single provider but we have no proof that all batches of tincture used in our experiments were from the same harvest. The ethanol used for menstruum was sugar beet alcohol.

The manufacturer believes bottles of tincture have a shelf-life of three years. All our experiments were conducted with tincture less than six months old and we always used tincture from a single batch for experiments in which comparisons were made.

Tincture spun at 4,000 RPMs in a tabletop centrifuge (rcf: 1790Xg) for 5 minutes produced a grey pellet and a red-brown, transparent (clear) supernatant. Pellets resuspended in water, did not produce a persistent foam when shaken, however supernatants retained their ability to foam. This is evidence that saponins are dissolved in the supernatant (as expected).

Fifteen millilitres from each batch of tincture and supernatant of the tinctures were dried in pre-weighed containers and their mass measured. Table 2 shows that, on average, one millilitre of supernatant had a dry mass of 38.2 (±2.9) mg.

**Table 2. Batch quality of four batches of tinctures.**

| BATCH | pH | specific gravity | ethanol | tincture | supernatant | pellet |
|---|---|---|---|---|---|---|
| | | | | (mg/ml) | | |
| #33047 | 5.7 | 0.9301 | 43% | 34.7 | 33.5 | 1.2 |
| #37769 | 5.7 | 0.9321 | 42% | 41.8 | 40.2 | 1.6 |
| #45115 | 5.6 | 0.9359 | 40% | 37.9 | 37.7 | 0.2 |
| #50266 | 5.7 | 0.9433 | 36% | 39.7 | 38.7 | 1.0 |
| Mean | 5.68 | 0.9354 | 40% | 39.9 (±3.1) | 38.2 (±2.9) | 1.1 (±0.6) |

Of the 120 g of marc mixed with each litre of menstruum, (120 g /L = 120 mg/ml) only 32% (38.2 / 120 = 0.318) was retained in the tincture's supernatant, so the amount of menstruum-soluble molecules in tincture is enriched by 3-fold.

The specific gravity measurements indicated the tincture was 36–43% ethanol, instead of the prescribed 45%. This discrepancy may be due to loss of ethanol by evaporation during percolation or factors related to hydration of the marc.

## Hydrolysis of saponins in tincture

We used persistent foam assays to survey across a range of temperatures and acidities, to determine the combinations which reduced the amount of foam by at least half. This led us to investigate combinations of two temperatures, 70 $^o$C and 80 $^o$C, and two acidities, 2 N and 6 N HCl, across a series of time points. These were chloroform extracted to separate the sapogenins from unreacted saponin, the hydrophilic glycoside released by hydrolysis and other unwanted materials that are hydrophilic (lipophobic).

Each Reaction Mix was made by adding 12.5 ml of 12 N (concentrated) or 4 N hydrochloric acid (HCl) to a 50 ml polypropylene tube (commonly called "centrifuge tubes") containing 12.5 ml of tincture (batch# 37769), making 25 ml of Reaction Mix containing the components of the tincture at half their original concentration (e.g. 22.5% ethanol) in 2 N or 6 N HCl. Each tube was sealed tightly with its screw-cap lid and placed in a water bath at 70˚C or 80 $^o$C. After a specific length of time, each tube was transferred to float in a container of cold water to cool. Once cooled, a millilitre was transferred to a tube containing 9 ml of water to assay for persistent foam. The remaining 24 ml was transferred to a separatory funnel, extracted with 15 ml of chloroform and the extract sent for analysis.

We also made "Mock Reaction Mixes" to detect any background or endogenous sapogenins, by mixing equal volumes of water and tincture. With no further manipulations, this was transferred to a separatory funnel, chloroform extracted and sent for analysis.

## Mass spec

Samples were diluted 10:1 with a solution of ACN (HPLC grade) + 0.1% Trifluoroacetic acid to ensure favourable ionization conditions. 10 μL aliquots of each sample was directly injected onto a Shimadzu 2020 mass spectrometer equipped with an electro spray ionization source in scan mode (100 to 1000 M/Z). The interface temperature, DL temperature, and heat block temperature were held at 350 $^o$C, 200 $^o$C, and 200 $^o$C respectively. The detector voltage was held at 1.75 kV.

## qHNMR

Solvent was removed by rotoevaporation and solid remaining was weighed then dissolved in 1,4-bis(trimethylsilyl)benzene/deuterated chloroform (CDCl$_3$) solution (0.17 mg/mL). NMR spectra were collected on a Bruker AVANCE III-HD with a proton operating frequency of 500.13 MHz and a carbon-13 frequency of 125.77 MHz. $^1$H-NMR spectra were collected at 25˚C, using 32 scans, a 30˚ pulse (Bruker pulse program: zg30), and a relaxation delay of 5.0 seconds. 32k data points were collected and then zero fill to 64k post collection. The fid was then subject to a line broadening of 0.3 and the spectra calibrated based off the internal tetramethylsilane peak (0.00 ppm). Peak area was determined using qGSD in the processing software MestreNova v 14.1.0 with 5 fixed fitting cycles. $^1$H-$^{13}$C Hetrosingle quantum coherence (HSQC) was collected using non-uniform sampling at 25%. The spectrum was collected at 25˚ C, using 16 scans, the Bruker pulse sequence hsqcedetgpsisp2.3, and a relaxation delay of 1.5 seconds. 1024 data points were collected in the F2 dimension and 100 in the F1 dimension. Linear

prediction using the Zhu-Bax method was applied in the F2 dimension to raise the number of data points to 2048 and then sine squared 90° functions were applied to both the F1 and F2 dimension.

## NMR quantification methodology

Using the quantification internal standard, 1,4-bis(trimethylsilyl)benzene, the quantification of smilagenin and sarsasapogenin was performed. Protons attached to carbon 27 were used for quantification of the stereoisomers due to their unique chemical shifts (1.08 ppm sarsasapogenin: 0.79 ppm smilagenin). The longitudinal relaxation of these peaks was determined to be ~2.5 seconds using the inversion recovery method. Using the quantitative global standard deconvolution (qGSD) in the NMR processing software Mestrelab, the desired peak areas were quantified without any overlap contributions from impurities. Measurement of the internal standard methyl peaks was performed in the same method.

$$m_a = \frac{A_a}{A_{IS}} \cdot \frac{P_{IS}}{P_a} \cdot \frac{mw_a}{mw_{IS}} \cdot \frac{m_{IS}}{m_s} \qquad \text{[eq 1]}$$

Where $m_a$ is the mass fraction of the analyte in the examined sample, $A_a$ is the peak area of the analyte, $A_{IS}$ is the peak area of the internal standard, $P_a$ is the number of protons corresponding to the peaks of interest of the analyte, $P_{IS}$ is the number of protons corresponding to the peak of interest in the internal standard, $mw_a$ is the molecular weight of the analyte, $mw_{IS}$ is the molecular weight of the internal standard, $m_{IS}$ is the mass of internal standard in the sample, and $m_s$ is the mass of the sample being examined.

## Demonstration of oil solubility in solutions of ethanol and water

One millilitre of a 10% olive oil solution, made by mixing 1 ml of olive oil with 9 ml of ethanol, was transferred to vials containing 15mls of ethanol, water, or combinations of the two solvents, then mixed and photographed immediately.

## Results and discussion

### Maximum yield calculation

Yields of sapogenin (sarsasapogenin plus smilagenin) depend upon the amount of sarsasaponin in the tincture and the efficiency of hydrolysis.

The highest amount of saponin reported in *Smilax* is 3% (by dry mass, see Fig 3).

Each hydrolysed sarsasaponin (1048 g/mole) yields a mole of sarsasapogenin (417 g/mole).

Therefore, if the tincture is made with marc from sarsaparilla containing sarsasaponin in the highest reported concentration for saponins in the genus *Smilax* and all the saponin transfers into the menstruum and is hydrolysed, we should expect yields of no more than 1.40 g of sapogenin (sarsasapogenin plus smilagenin) from each litre of tincture hydrolysed.

(120g/L × 3% × 417 g/mole ÷ 1048 g /mole = 1.4g/L = 1.4 mg/ml)

### Endogenous sapogenins in tincture

The chloroform-extracted products from a 25 ml "Mock Reaction Mix" (to detect any background levels of endogenous sapogenins) found no sarsasapogenin or smilagenin. This experiment was repeated and gave the same negative results. A large scale "Mock Reaction Mix", containing 100 ml of tincture was chloroform extracted with 50 ml of chloroform. This was evaporated to dryness, resuspended in 10 ml of chloroform and sent for analysis. No

**Table 3. Results at four time points (top row) from four reaction conditions.**

| Reaction → | A: 6N HCl, 80 °C | | | | B: 2N HCl, 80 °C | | | | C: 6N HCl, 70 °C | | | | D: 2N HCl, 70 °C | | | |
|---|---|---|---|---|---|---|---|---|---|---|---|---|---|---|---|---|
| Incubation (hours) | 1 | 2 | **4** | 8 | 2 | 4 | **8** | 16 | 2 | 4 | **8** | 16 | 2 | 4 | **8** | 16 |
| Crude (mg/ml) | 2.33 | 2.86 | **3.85** | 3.29 | 3.36 | 3.31 | **3.36** | 3.50 | 5.08 | 5.32 | **5.16** | 5.76 | 3.19 | 3.19 | **3.34** | 4.18 |
| Sarsasapogenin (mg/ml) | 0.34 | 0.47 | **0.65** | 0.46 | 0.57 | 0.96 | **1.09** | 1.05 | 0.37 | 0.42 | **0.44** | 0.23 | 0.37 | 0.71 | **0.90** | 0.76 |
| Smilagenin (mg/ml) | 0.05 | 0.08 | **0.13** | 0.11 | 0.10 | 0.20 | **0.23** | 0.21 | 0.21 | 0.22 | **0.25** | 0.18 | 0.04 | 0.08 | **0.13** | 0.12 |
| Sapogenin (mg/ml) | 0.39 | 0.55 | **0.78** | 0.57 | 0.66 | 1.16 | **1.32** | 1.27 | 0.58 | 0.64 | **0.69** | 0.41 | 0.40 | 0.79 | **1.03** | 0.88 |
| Sapogenins/Crude | 17% | 19% | **20%** | 17% | 20% | 35% | **39%** | 36% | 11% | 12% | **13%** | 7% | 13% | 25% | **31%** | 21% |
| Saponins (mg/ml) | 0.97 | 1.38 | **1.97** | 1.43 | 1.67 | 2.93 | **3.32** | 3.19 | 1.47 | 1.61 | **1.75** | 1.03 | 1.02 | 1.99 | **2.59** | 2.22 |
| Saponins / marc | 0.8% | 1.1% | **1.6%** | 1.2% | 1.4% | 2.4% | **2.8%** | 2.7% | 1.2% | 1.3% | **1.5%** | 0.9% | 0.8% | 1.7% | **2.2%** | 1.9% |
| Smilagenin / Sapogenin | 13% | 15% | **16%** | 20% | 15% | 17% | **18%** | 17% | 36% | 35% | **37%** | 44% | 9% | 10% | **12%** | 13% |

sarsasapogenin or smilagenin was detected by qNMR of a 5 ml sample (representing the chloroform-extractable fraction from 50 ml of tincture).

## Hydrolysis of tincture saponins

Hydrolysis is expected to increase with increasing acidity, temperature and duration of exposure (incubation time). However, we were concerned that excessive temperatures or acidity could decompose our desired product, so designed our experiments to identify incubation times where yields decreased. The time point previous to the one showing the decrease was deemed "optimal" (although a time point between these two is likely to be "more optimal").

Tincture was incubated in equal volumes of 4N or 12 N HCl at 70 °C or 80 °C for different lengths of time. Each of the 25 ml Reaction Mixtures was extracted once with 15 ml of chloroform and the recovered chloroform was sent for analysis of this, 5 ml was used to produce the data in Table 3, which displays the optimal time for each experiment in bold. (An explanation of the values follows the table.).

## Data (Products)

**Crude** (mg/ml) is the dry mass (mg) of the chloroform-extractable materials produced per millilitre of tincture (ml). This is a gravimetric value (determined by weight).

**Sarsasapogenin** (mg/ml) is the mass (mg) of sarsasapogenin per millilitre (ml) of tincture. This was determined by qNMR.

**Smilagenin** (mg/ml) is the mass (mg) of smilagenin per millilitre (ml) of tincture. This was determined by qNMR.

## Calculations

**Sapogenin** (mg/ml) is the sum of sarsasapogenin and smilagenin mass (mg) per millilitre of tincture (ml). This is a measure of hydrolysis.

**Smilagenin / Sapogenin** (%) is smilagenin (mg/ml) divided by total sapogenin (mg/ml). This is a measure of epimerisation.

**Sapogenins / Crude** (%) is the sapogenins per millilitre (mg/ml) divided the chloroform-extractable product (mg/ml). This is a measure of product purity.

**Saponins** (mg/ml) is the mass of saponins (mg) hydrolysed per millilitre of tincture (ml). It is calculated by multiplying the amount of sapogenin by 2.52. (saponins = 1048 g / mol, sapogenins = 416g /mol ratio = 2.52).

**Saponins (mg/ml)/marc** is the saponins hydrolysed (mg/ml) per marc (mg/ml), with marc = 120mg/ml. It reflects the contribution of hydrolysable saponins make to the mass of the marc (which was 3% in Fig 3).

For our first experiment we incubated tincture at 80 °C, because this temperature was the highest among the protocols discussed. We decided to test an acidity approximately three times greater than Jacob and Simpson reported (50% greater than Wells *et al.* recommended) to compensate for the lack of refluxing. The results are in Table 3(A) and displayed graphically in Fig 6 (left side).

The maximum yield of 0.78 mg of sapogenins (0.65 mg of sarsasapogenin and 0.13 mg of smilagenin) per millilitre of tincture was achieved after four hours. This is 56% of the maximum yield possible (0.78/1.4), calculated earlier.

Doubling the incubation time to eight hours reduced the yield (by 27% from 0.78 mg/ml to 0.57 mg/ml; 0.57/0.78 = 0.73), indicating that excessive exposure to 6N HCl at 80 °C causes decomposition of our desired products.

The dry mass of the crude chloroform-extracted product from the four-hour incubation was 3.85 mg/ml, which means that final product was 20% sapogenins (0.78 mg/3.85 mg = 0.20).

The results from this experiment (A) convinced us that hours of refluxing are not required for hydrolysis of sarsasaponin in sarsaparilla tincture.

In our second experiment, we lowered the acidity from 6N to 2N, hoping to minimize degradation. The results are in Table 3B and displayed graphically in Fig 6 (right side).

At the lower acidity, the rate of hydrolysis decreased (note that in the second experiment, incubation times start doubling from hour 2 not hour 1) and the maximum yield increased, to 1.32 mg of sapogenins per ml of tincture. This is 94% (1.32/1.40) of the maximum theoretical yield.

The crude is 39% sapogenins after 8 hours at 80 °C, in 2N HCl.

Doubling incubation time for the highest yield reaction from 8 hours to 16 hours reduced the total amount of sapogenins from 1.32 to 1.27, a loss of only 4% (1.27/1.32 = 0.96).

The ratios of smilagenin to sapogenin were similar regardless of normality or duration of the reactions.

The results from this experiment (B) convinced us that hydrolysis and degradation proceed more slowly in 2N HCl, yielding more of the desired products (sapogenins) and less undesired

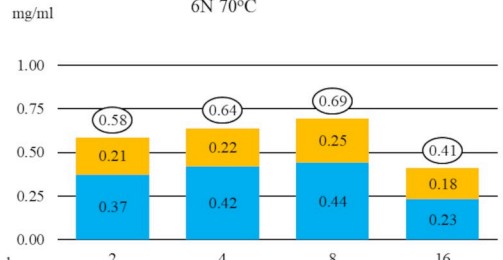
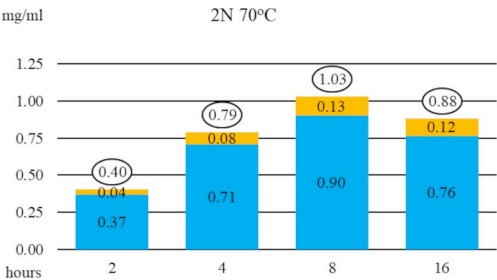

**Fig 6. Yields from tincture incubated at 80 °C in 6N HCl for 1, 2, 4 and 8 hours (left) and in 2N HCl for 2, 4, 8 and 16 hours (right).** Colours correspond to sarsasapogenin (blue), smilagenin (orange) and non-sapogenins (grey). The circled values above each column are the total sapogenins (sum of sarsasapogenin and smilagenin).

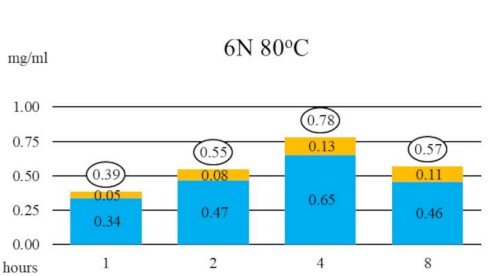
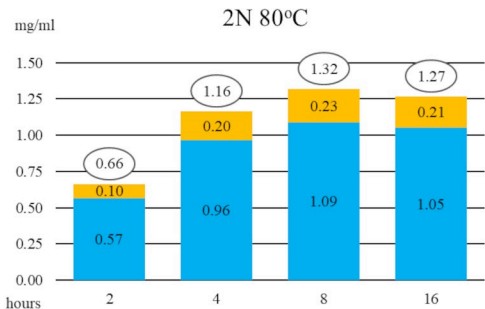

**Fig 7. Yields from tincture incubated at 70 °C in 6N HCl (left) and 2N HCl (right) for 2, 4, 8 and 16 hours.** Colour coded as in Fig 6.

products (degraded sapogenins) than in 6N HCl. Doubling the incubation time caused only a small change in yields, suggesting that reactions in 2N HCl are easier to control than those in 6N HCl.

These small-scale experiments provided adequate quantities of product for our analytical work, but our goal was production of smilagenin and sarsasapogenin in large quantities. We scaled up by incubating 100 ml of HCl with 100 ml of tincture in a 250 ml high-density polyethylene (HDPE) bottle with polypropylene closure (screw lid). However, these larger reaction vessels, randomly breached—rapidly ejecting copious amounts of hot acidic persistent foam. This is because ethanol's boiling point is 78 °C (at one atmosphere), so as the reaction warmed up to 80 °C, the ethanol vaporised, causing the pressure inside the container to soar.

Therefore, we repeated the previous experiments at 70 °C. The results are in Table 3 (C and D) and displayed graphically here in Fig 7.

Lowering the temperature of the 6N reaction from 80 °C to 70 °C, doubled the time to the maximum yield of sapogenin (from for 4 hours to 8 hours), decreased the maximum yield of sapogenins by 12% (from 0.78 to 0.69; 0.69 / 0.78 = 0.88), and decreased purity (from 20% to 13%). Curiously, the 6N HCL reaction produced approximately twice as much smilagenin at the lower temperature. We do not know why.

Lowering the temperature of the 2N reaction from 80 °C to 70 °C, decreased the maximum yield of sapogenins (from 1.32 to 1.03) and decreased purity (from 39% to 31%).

Table 4 summarises the results of our hydrolysis experiments.

## Chloroform extractions of saponins and sapogenins

The intact saponins in unhydrolyzed tincture (from Mock Reactions, to detect endogenous sapogenins in tincture) formed an opaque emulsion of micelles when mixed with chloroform. See Fig 8.

**Table 4. Changes in maximum yields at 80 °C or 70 °C and 6N or 2N HCl.**

|  | 6N | → lower acidity → | 2N |
|---|---|---|---|
| 80 °C | 4 hours to max 0.78 mg/ml sapogenins 12% pure | twice as much time to max 70% more sapogenin nearly quadrupled purity | 8 hours to max 1.32mg/ml sapogenins 47% pure |
| → lower temperature → | twice as much time to max 45% more sapogenin more than tripled purity |  | same time to max 25% less sapogenin decreased purity |
| 70 °C | 8 hours to max 1.13mg/ml sapogenins 44% pure | took the same time to max Produce slightly less sapogenins and slightly lower purity | 8 hours to max 1.03mg/ml sapogenins 37% pure |

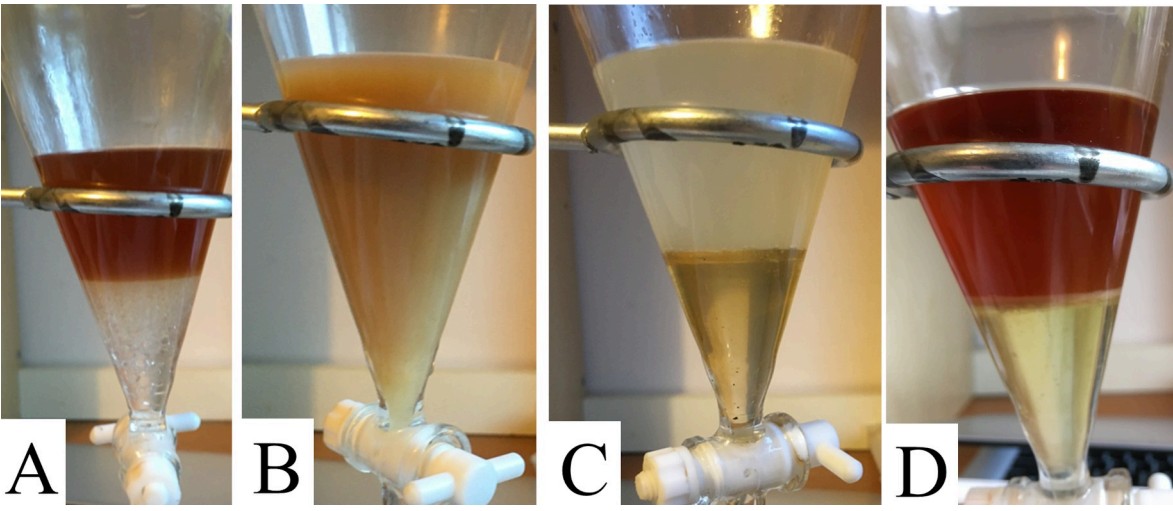

**Fig 8. Photographs of separatory funnel containing chloroform and unhydrolyzed tincture.** A) Chloroform gently added to Mock Reaction Mixture settled below it. B) When mixed, an emulsion formed. C) An hour later, most of the chloroform, now tinted, had returned to the bottom and the emulsion lost its colour. D) Several hours later, after the emulsion had dispersed, the chloroform was collected, leaving this aqueous layer containing saponins.

Within an hour, the chloroform settled to the bottom, carry a light-yellow tint. The emulsion lost much of its colour, causing it to appear white due to light scattering. Fig 8C.

Several hours later, the emulsion had completely dispersed, and the yellow-tinted chloroform collected, leaving behind the aqueous layer, now seen by light transmitted through the solution and tinted red. Fig 8D. This chloroform-extracted tincture produced the same amount of persistent foam as it had before extraction, proving the saponins remained in the aqous layer.

Hydrolysed tincture was easier to extract. The aqueous and organic (chloroform) layers separated quickly and distinctly. See Fig 9A The chloroform layer differed from the chloroform collected from unhydrolyzed tincture in that it appeared cloudier and slightly orange or brown.

The aqueous layer contained a "dark disc" that formed at the bottom of the aqueous layer, just above the chloroform. This disk usually broke away and floated to the top of the aqueous layer and always adhered to the vessel as the chloroform was collected.

The aqueous layer looked unaffected by the extraction, other than the appearance of a dark disk, but it was now unable to create a persistent foam, proving the saponins were now gone.

A dark disk was retrieved from the aqueous layer remaining after chloroform extraction of 200 ml of Reaction Mix, made from 100 ml of tincture. When dried, the black material was friable and had a mass of 290 mg, so 2.9 mg was produced from each millilitre of hydrolysed tincture.

We believe that the dark disk is composed mostly of the glycoside (glycone) released by the acid hydrolysis of sarsasaponin. This hydrophilic molecule has a molecular weight of 631g/mol, so 2.2 mg are expected from each millilitre of hydrolysed tincture.

$$(120g/L \times 3\% \times 631 \text{ g/mole} \div 1048 \text{ g /mole} = g/L = 2.2 \text{ mg/ml})$$

## Comparison of the methods

Our method for producing sapogenins from *Smilax ornata* root powder differs from previously reported methods, starting with the first step; preparing the saponin for hydrolysis. Fig 10 compares the two methods.

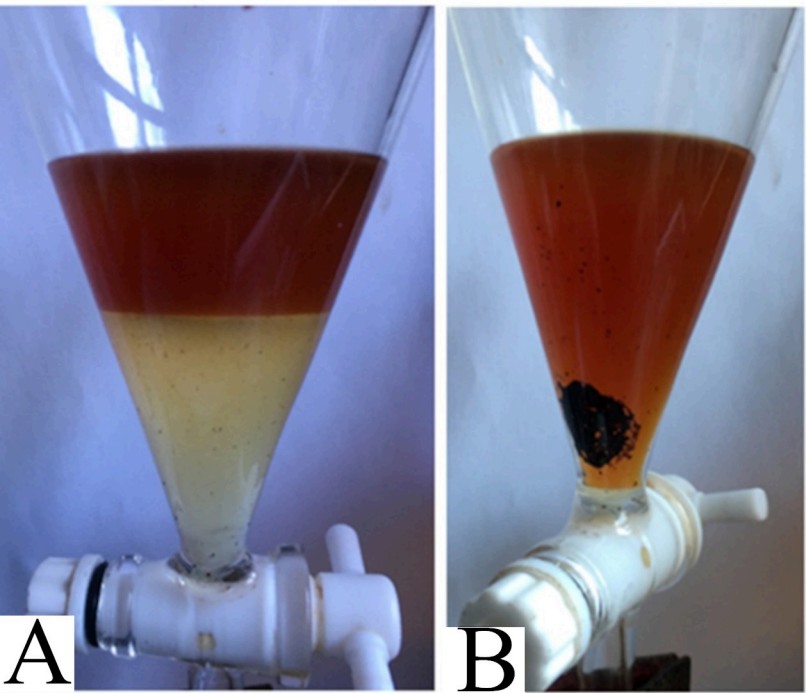

**Fig 9. Photographs of separatory funnel containing chloroform and hydrolysed tincture.** A) Aqueous (top) and chloroform (bottom) layers, ready to be collected. B) Aqueous layer remaining after chloroform layer has been collected, displaying "dark disk" that forms during extraction of hydrolysed tincture.

Askew, Farmer and Kon followed the instructions of Jacobs and Simpson for preparation of the saponin and its hydrolysis. Both teams started by percolating 95% denatured ethanol through the powder, trapping the saponin in "a brown viscous gum which weighed 32.2 kilos", quoting Jacobs & Simpson, composed partly of fat. They "defatted" the gum by dissolving it in 85% alcohol, then extracted it with ligroin at 50–60 °C, three times. At this point Jacobs & Simpson reported, "The total yield was 26.9 kilos of fat-free gum". If 3% of the powder is saponin, and all of it transferred to the gum, 6.75 kg of saponin (3% of 225 kg) are in the 26.9kg fat-free gum, making the gum 25% saponin (6.75 / 26.9).

Our tincture uses 480 g of root powder to make 4 L of tincture (120 g/L), that have, on average, 38.2 mg of dissolved solids per millilitre (Table 2), so only 32% of the marc went into solution (38.2 mg / 120 mg = 0.318). Assuming, as we did above, saponin constitutes 3% of the powder and all of it migrated into the solution, 3.6 mg (120 mg / ml X 3% = 3.6) of that 38.2 mg of dissolved solids in the tincture should be saponin, making the tincture 9% (3.6/ 38.2 = 0.094) saponin.

The gum has has almost three times as much saponin as the tincture (25% / 9% = 2.77) so appears to be a superior feedstock for hydrolysis. However, the composition of the non-saponin fraction of the gum, ~20.15 kg of the 29.6 kg of gum (or 75%), might include up to 5.6 kg of resin (2.5% of 225 kg, see Fig 3) and nonspecific materials caught in the gum including long polymers such as cellulose, starch and DNA, that would contribute to the gum's viscous property and interfere with subsequent reactions.

The methods for hydrolysis and purification of the sapogenins are very different, so a one-to-one comparison is, admittedly, flawed. That said, Fig 11 continues to compare the protocols from the point where Fig 10 ended.

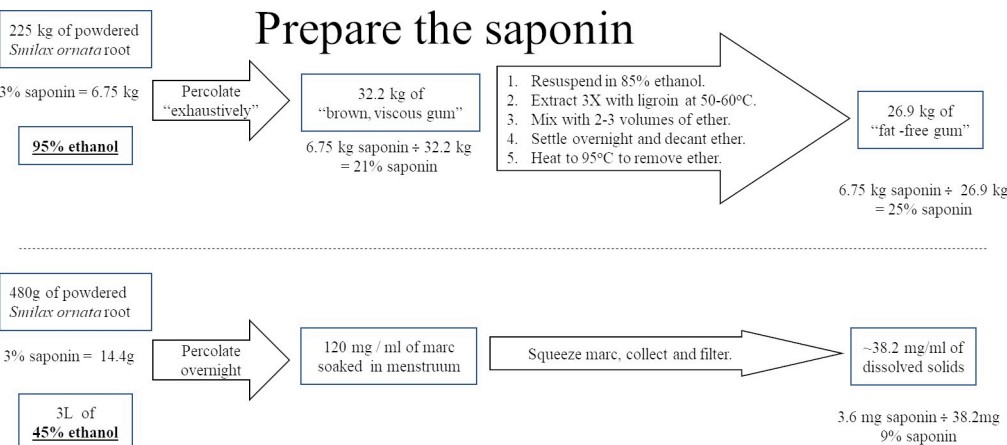

**Fig 10. Comparison of methods for saponin preparation.** Top diagram details the protocol used by Jacob and Simpson, with the values they reported. Bottom diagram displays our protocol for comparison.

Our tincture is prepared by soaking the plant materials in "menstruum"–an aqueous solution of 45% ethanol. Approximately 40% of the plant material transfers into the menstruum (see Table 2) apparently leaving behind the "gum' precursors in the 60% that stays in the marc.

We believe that fat solubility is responsible for the difference in outcomes. Lipids, being non-polar organic compounds, are soluble in organic solvents such as ethanol, but insoluble in water. Lipids dissolved in a solution of ethanol spontaneously comes out of solution when added to water, by forming micelles. [19] A mixture of ethanol and water has an intermediate affect, as demonstrated in Fig 12.

This demonstration used olive oil, but *Smilax ornata* root powder would be expected to contain different kinds of lipids and could create a variety of micelles differing in size, stability, charge, *etc*. Regardless, a population of lipids will display the same general trend—they will dissolve in ethanol or a solution high in ethanol, but in a more aqueous solution they will form micelles. The hydrophobic/hydrophilic interactions in a solution of ethanol or water are relatively easy to predict, but in menstruum (45% ethanol 55% water) those interactions are likely to be complicated, dynamic and may include formation of reverse micelles.

We believe that, when tincture is made, micelles form from lipids in the marc, precipitate from solution and constitute part of the waste left behind when the tincture is collected. The

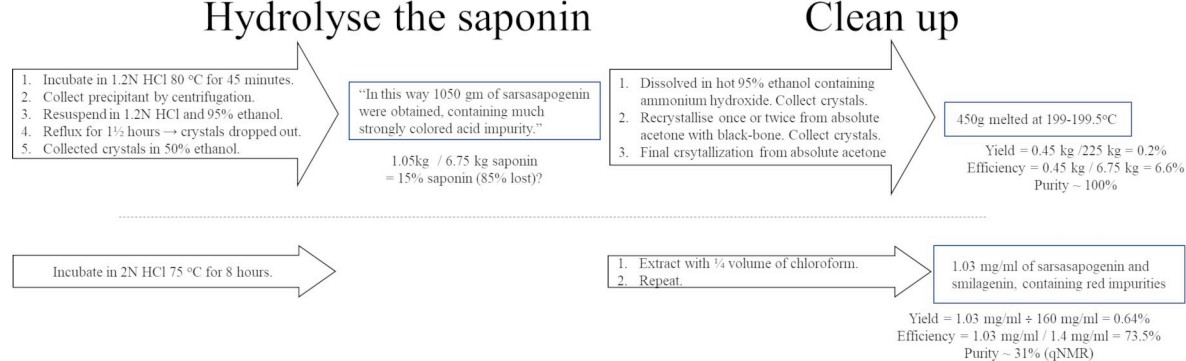

**Fig 11. Comparison of methods for hydrolysis and collection of sapogenins.** Top diagram details the protocol used by Jacob and Simpson, with the values they reported. Bottom diagram displays our protocol for comparison.

**Fig 12. Demonstration of oil solubility in solutions of ethanol and water.** The photograph on the left was taken by ambient light and the photograph on the right was taken with the assistance of a flash.

micelles may also carry and sequester, within them and on them, other molecules, such as resins, which would otherwise contribute to the fat-free gum.

## Conclusion

For nearly a century, saponins have been "prepared" for hydrolysis starting with a counterproductive step that imbedded the saponins in a viscous gum, confounding subsequent steps and that could only be overcome by the use of large amounts of effort, organic solvents and heat. Hydrolysis of saponins in tincture provides a simple, inexpensive and environmentally friendly alternative.

## Author Contributions

**Conceptualization:** Jamie Love.

**Investigation:** Jamie Love, Casey R. Simons.

**Validation:** Casey R. Simons.

**Writing – original draft:** Jamie Love.

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
