## [Decision Letter · Decision Letter 0]

20 Nov 2020

PONE-D-20-31354

Acid hydrolysis of saponins extracted in tincture

PLOS ONE

Dear Dr. Love,

Thank you for submitting your manuscript to PLOS ONE. After careful consideration, we feel that it has merit but does not fully meet PLOS ONE’s publication criteria as it currently stands. Therefore, we invite you to submit a revised version of the manuscript that addresses the points raised during the review process.

We look forward to receiving your revised manuscript.

Kind regards,

Pasquale Avino, Ph.D.

Academic Editor

PLOS ONE

'The author(s) received no specific funding for this work.'

We note that one or more of the authors are employed by a commercial company: Synapses Ltd, Mill of Fyall.

Reviewers' comments:

Reviewer #1: This manuscript describes attempts to improve sapogenin hydrolysis and purification from hydroalcoholic extracts of Smilax. The use of qNMR is innovative and provides a method that can be repeated easily in different While they have made progress to that end there are some issues that need to be addressed.

1) For any phytochemical study the plant identity needs to be confirmed. Usually this is done with a herbarium voucher that is identified by a taxonomist. At first mention, the botanical name is given with taxonomic authority and family in brackets Similar ornate Hook. (Smilacaceae). Thereafter it is referred to as S. ornata.

2) Major findings are mostly yields under different conditions (ex table 2,3 and figures 6and 7). These values need to be presented as a mean and standard error with number of replicates. Ideally a multiple comparison test would show which methods are statistically superior. For this reason also pie charts are rarely accepted in peer reviewed literature

3) The paper reads like a thesis rather than a focussed scientific paper with concise introduction, methods results and discussion. For example.

a. The introduction is long and rambling. Avoid point form and subsections, quotations.

b. Methods are presented in results

c. Equations are not numbered

d. Convert machine specific rpm to universal unit xg

e. Ensure that appropriate units in brackets are placed above columns

---

## [Author Response · Author response to Decision Letter 0]

4 Dec 2020

EDITORS

RESPONSE TO REVIEWERS

1. Please ensure that your manuscript meets PLOS ONE's style requirements.

RESPONSE: 

I believe we meet the requirements but if you believe we have not, please specify.

2. In your Data Availability statement, you have not specified where the minimal data set underlying the results described in your manuscript can be found. 

RESPONSE: 

All our data are found in the tables (Table 2 and Table 3) in the manuscript. This is not a data heavy report.

'The author(s) received no specific funding for this work.'

We note that one or more of the authors are employed by a commercial company: Synapses Ltd, Mill of Fyall.

a, b and c

RESPONSE: 

Synapses is not a commercial company. 

A couple years ago I retired (from U of Edinburgh) but have not retired from research. I have a home lab (shed) in which I conduct all my work, including the work that contributed to this manuscript. As a “civilian” (no longer with the university), I had difficulty purchasing certain chemicals – chloroform, for example. Sigma, et al. will not ship to private individuals or residential addresses. Synapses, was the name I used as a consultant (independent of the university) since 1998, so I decided to incorporate it – and now I can order the chemicals I need.

Synapses is my “hobby lab” where I tinker with projects that interest me. Synapses is non-profit and totally self-funded and I am currently unincorporating it.

I have no competing or conflicting interest. I will not (cannot) profit from publishing “Acid hydrolysis of saponins extracted in tincture” in PLOS ONE. 

“This does not alter our adherence to PLOS ONE policies on sharing data and materials.”

I would be happy to address any additional concerns.

Reviewers' comments:

Reviewer #1: This manuscript describes attempts to improve sapogenin hydrolysis and purification from hydroalcoholic extracts of Smilax. The use of qNMR is innovative and provides a method that can be repeated easily in different While they have made progress to that end there are some issues that need to be addressed.

1) – a - For any phytochemical study the plant identity needs to be confirmed. Usually this is done with a herbarium voucher that is identified by a taxonomist. 

RESPONSE:

We are not trying to prove that sarsasapogenin is present in S. ornata. That was done in 1934 by Jacob and Simpson (and referenced). Nor are we trying to prove that smilagenin is produced from S. ornata. That was done in 1936 by Askew, Farmer and Kon. These observations have been confirmed since their first discovery by several investigators (including RE Marker, a luminary in steroidal chemistry).

That said, I understand and appreciate the reviewer’s point. Proper identification is crucial and an herbal voucher is preferred. However, we do not have the resources to collect the specimens (in Central America) nor the skill to identify them. Instead, we must trust the supplier of the tincture to have used the correct plant. Our tincture was prepared by and purchased from a trusted, experienced and long-established local herbalist who knows how to source the marc (plant material) and judge its quality – important skills that we lack. 

BTW – sarsasapogenin has been reported to be in the roots of Smilax lanceolata and S. rotundifolia, Dioscorea colletii (a yam), nineteen species of yucca (Yucca spp.), Anemarrhena asphodeloide (used in traditional Chinese medicine) and several species of asparagus (Asparagus spp) including the domesticated species, as well as the seeds and leaves of Trigonella foenum-graecum. We are collecting powder made from all these plants (for a follow up paper) but, again, must rely upon the suppliers for identification of most of them.

1) -b -At first mention, the botanical name is given with taxonomic authority and family in brackets Hook. (Smilacaceae). Thereafter it is referred to as S. ornata.

RESPONSE: 

My mistake. I’ve added the authority and family at first mention and corrected the spelling errors (ornata to ornate). 

Thank you.

2) -a -Major findings are mostly yields under different conditions (ex table 2,3 and figures 6and 7). These values need to be presented as a mean and standard error with number of replicates. 

RESPONSE: 

Table 2 shows the raw data, mean and standard deviation from four batches of tincture. Our intention in this table is not to find the average (mean) of each batch but the average across four batches. 

Table 3 shows the raw data with (only) a single data point for each of the 16 experiments. We did not run replicates because the volume of tincture needed to run (say four) replicates of each point would have exceeded the amount of tincture in a single batch. We wanted to make comparisons across all four conditions (combinations of temperature and acidity) at four different timepoints using the same (single) batch of tincture. The raison d'être for Table 3 is not accuracy (which would require replicates) but in comparisons between the conditions. We believe that others who replicate our Table 3 in their own labs using a single batch of S. ornata tincture, will come to similar conclusions about the relationship of between the conditions, but unlikely to get the same values.

2) - b - Ideally a multiple comparison test would show which methods are statistically superior. 

RESPONSE:

We are not trying to find the statistically superior method. We are using the data in Table 3 to display trends. Importantly, we are not getting across our point and apologize for the confusion. Therefore, we have included a new table (Table 4. Diagram illustrating changes in maximum yields at 80 oC or 70 oC and 6N or 2N HCl) to highlight the conclusions that others may use to guide them in finding conditions that work with the saponin of their interest.

2) – c - For this reason also pie charts are rarely accepted in peer reviewed literature

RESPONSE:

The pie charts have been removed.

3) The paper reads like a thesis rather than a focussed scientific paper with concise introduction, methods results and discussion. For example.

RESPONSE:

It felt like a thesis too! We are working across several specialties, so try to be complete but has been trimmed and specific concerns are addressed below.

a. The introduction is long and rambling. Avoid point form and subsections, quotations.

RESPONSE:

Edited to be succinct and removed some parts.

Points and quotes are removed and subsections limited. I hope it reads better.

b. Methods are presented in results

RESPONSE:

I assume the reviewer is referring to the demonstration of the oil in different concentrations of alcohol. I’ve moved the method, but not the result, into the Methods section.

c. Equations are not numbered

RESPONSE:

The equation for qNMR is now numbered [eq 1] and Imassume that is what the reviewer had in mind. (Elsewhere, I show my maths to make calculations, but I believe they do not need to be numbered (and would look awkward if I did). I hope that is satisfactory.

d. Convert machine specific rpm to universal unit xg

RESPONSE:

Added.

e. Ensure that appropriate units in brackets are placed above columns

RESPONSE:

I don’t see an error. Can you be specific?

ALSO:

I felt that the data and discussion about background was poorly explained and upon reflection, decided it added nothing of value to the paper – so I removed the data from Table 3 (the last three rows) about the background crude and its discussion later on. 

I found and corrected a few typos.

Thank you.

Jamie

---

## [Decision Letter · Decision Letter 1]

15 Dec 2020

Acid hydrolysis of saponins extracted in tincture

PONE-D-20-31354R1

Dear Dr. Love,

We’re pleased to inform you that your manuscript has been judged scientifically suitable for publication and will be formally accepted for publication once it meets all outstanding technical requirements.

Kind regards,

Pasquale Avino, Ph.D.

Academic Editor

PLOS ONE

Reviewers' comments:

Reviewer #1: The authors have responded positively to the review with revisions and explanations. The manuscript is acceptable for publication.

---

## [Editor Report · Acceptance letter]

18 Dec 2020

PONE-D-20-31354R1 

Acid hydrolysis of saponins extracted in tincture 

Dear Dr. Love:

I'm pleased to inform you that your manuscript has been deemed suitable for publication in PLOS ONE. Congratulations! Your manuscript is now with our production department. 

Kind regards, 

on behalf of

Professor Pasquale Avino 

Academic Editor

PLOS ONE